# Suppression of Age-Related Macular Degeneration-like Pathology by c-Jun N-Terminal Kinase Inhibitor IQ-1S

**DOI:** 10.3390/biomedicines11020395

**Published:** 2023-01-29

**Authors:** Anna A. Zhdankina, Dmitry I. Tikhonov, Sergey V. Logvinov, Mark B. Plotnikov, Andrei I. Khlebnikov, Nataliya G. Kolosova

**Affiliations:** 1Department of Histology, Embryology and Cytology, Siberian State Medical University, 634055 Tomsk, Russia; 2Department of Pharmacology, Goldberg Research Institute of Pharmacology and Regenerative Medicine, Tomsk NRMC, 3 Lenin Ave., 634028 Tomsk, Russia; 3Radoiphysical Faculty, National Research Tomsk State University, 634050 Tomsk, Russia; 4Kizhner Research Center, Tomsk Polytechnic University, 30 Lenin Ave., 634050 Tomsk, Russia; 5Federal Research Center Institute of Cytology and Genetics SB RAS, Pr. Lavrentiev, 10, 630090 Novosibirsk, Russia

**Keywords:** age-related macular degeneration, c-Jun N-Terminal Kinase Inhibitor IQ-1S, OXYS rats

## Abstract

Age-related macular degeneration (AMD) is the leading cause of irreversible visual impairment worldwide. The development of AMD is associated with inflammation, oxidative stress, and progressive proteostasis imbalance, in the regulation of which c-Jun N-terminal kinases (JNK) play a crucial role. JNK inhibition is discussed as an alternative way for prevention and treatment of AMD and other neurodegenerative diseases. Here we assess the retinoprotective potential of the recently synthesized JNK inhibitor 11*H*-indeno[1,2-*b*]quinoxalin-11-one oxime sodium salt (IQ-1S) using senescence-accelerated OXYS rats as a model of AMD. The treatment with IQ-1S (50 mg/kg body weight intragastric) during the period of active disease development (from 4.5 to 6 months of age) improved some (but not all) histological abnormalities associated with retinopathy. IQ-1S improved blood circulation, increased the functional activity of the retinal pigment epithelium, reduced the VEGF expression in the endothelial cells, and increased the expression of PEDF in the neuroretina. The result was a decrease in the degeneration of photoreceptors and neurons of the inner layers. IQ-1S significantly improved the retinal ultrastructure and increased the number of mitochondria, which were significantly reduced in the neuroretina of OXYS rats compared to Wistar rats. It seems probable that using IQ-1S can be a good prophylactic strategy to treat AMD.

## 1. Introduction

Age-related macular degeneration (AMD) is the leading cause of irreversible visual impairment and blindness in industrialized countries, and is defined as a chronic, multifactorial, and progressive central retinal disease. The prevalence of AMD is increasing dramatically as the proportion of the elderly in the population continues to rise [1]. AMD is a multifactorial disease involving a complex interplay of genetic, environmental, metabolic, and functional factors. Clinically, AMD is classified into dry (atrophic) AMD and wet (exudative) AMD depending on the presence of choroidal neovascularization. Most AMD starts as the dry type, and in 10–20% of individuals it progresses to the wet type. Although the introduction of anti-angiogenesis therapy has helped to prevent blindness and restore vision in wet AMD, there remains no effective treatment for the patients with dry (~90% of all cases) AMD [2]. The dry form is characterized by progressive degeneration of the retinal pigment epithelium (RPE) and photoreceptor cells that is due to age-related changes in the retina [3,4]. The etiology of AMD remains largely unknown, but it has been suggested that an equilibrium between proangiogenic vascular endothelial growth factor (VEGF) and antiangiogenic neurotrophic pigment epithelium-derived factor (PEDF) is important for regulation of retinal physiology and for pathophysiology, including late AMD [5,6,7]. Anti-VEGF treatment has turned out to be a good way to control neovascular wet AMD more effectively, but there is no effective treatment for the dry form.

It is also obvious that the development of this complex degenerative disease is associated with inflammation, oxidative stress, and progressive proteostasis imbalance in the endoplasmic reticulum (ER stress). Representatives of the family of stress-induced, mitogen-activated protein kinases (MAPs) play an important role in the processes of inflammation and the production of cytokines, apoptosis, neurodegeneration, proliferation, and the differentiation of cells. Among them are c-Jun N-terminal kinases (JNK), which modulate various cellular processes, including cell proliferation, apoptosis, autophagy, and inflammation. Three different genes encode three isoforms of the protein: JNK1, JNK2, and JNK3. JNK1 and JNK2 are expressed ubiquitously throughout the body, while JNK3 expression is seen mainly in the brain, heart, and testicles [8]. There is strong evidence that alterations in JNK signaling play a crucial role in the pathogenesis of neurodegenerative diseases, because the sustained activation of JNK leads to synaptic dysfunction and even neuronal apoptosis. Violations of signaling in JNK-controlled pathways are observed in the development of various neurodegenerative diseases [9]. As studies on various retinal cell cultures and animal models indicate, JNK signaling may contribute to the pathogenesis of AMD [10]. This is indicated by the fact that mice lacking JNK1 exhibit decreased inflammation, reduced choroidal neovascularization (CNV), lower levels of choroidal VEGF, and impaired choroidal macrophage recruitment in a murine model of wet AMD [11]. Wherein hypoxia-induced JNK1 activation promotes retinal VEGF production and pathological angiogenesis in a murine model of retinopathy of prematurity [12].

It is natural that pharmacological JNK inhibition is discussed as an alternative avenue for prevention and treatment of AMD as well as other neurodegenerative diseases [13]. To date, a number of non-protein synthetic JNK inhibitors have been described, including SP600125, AS601245, IQ-1S, SR-3306, and SU3327, as well as protein and non-protein molecules that inhibit JNK signaling [7,14,15,16,17,18,19]. Some of them have demonstrated neuroprotective activity in animal models of stroke [20,21,22,23,24]. Among them is the recently synthesized JNK inhibitor 11*H*-indeno[1,2-*b*]quinoxalin-11-one oxime sodium salt (IQ-1S), whose ability to inhibit JNK has been shown in silico [19,25,26], and neuroprotective properties were demonstrated in models of cerebral ischemia [20,27,28].

The aim of this study was to evaluate the retinoprotective potential of IQ-1S using senescence-accelerated OXYS rats, characterized by developing of a retinopathy that by the clinical, morphological, and ultrastructural signs corresponds to AMD in humans [29,30,31]. The first clinical manifestations of retinopathy are detected during ophthalmoscopic examinations in ~20% of OXYS rats at 5–6 weeks of age, and at the age of 3–4 months signs of retinopathy are recorded in all animals. Pathological changes progress and reach pronounced stages, suggesting loss or significant deterioration in visual acuity, by the age of 14–18 months. Retinopathy that develops in OXYS rats corresponds to the “dry” form of AMD, and is manifested by the signs for this disease: dystrophic changes and thinning of the retina, impaired microcirculation in the choroid, changes in neurotrophic supply, accumulation of lipofuscin and amyloid β, and structural characteristics of AMD [7,14,32]. At the same time, neovascularization develops with age in some (∼10–20%) OXYS rats, like in humans. To assess the therapeutic potential of IQ-1S, we used OXYS rats aged 4.5 months. As we showed earlier, at this age OXYS rats exhibit the pronounced signs corresponding to the first stage of the disease (according to the Age-Related Eye Disease Study grade protocol—http://eyephoto.ophth.wisc.edu (acessed on 28 November 2022)): appearance of drusen and other pathological changes in the retinal pigmented epithelium (RPE), and partial atrophy of the choroid capillary layer [33]. Here we examined the effect of IQ-1S on the AMD-like pathology development based on a structural analysis of the retina and expression of VEGF and PEDF.

## 2. Materials and Methods

### 2.1. Animals

The OXYS rat strain was developed at the Institute of Cytology and Genetics (ICG), SB RAS (Novosibirsk, Russia), from a Wistar stock as described earlier [34]. Males of 4-month-old OXYS rats, and age-matched Wistar rats, were obtained from the Breeding Experimental Animal Laboratory of the ICG SB RAS (Novosibirsk, Russia). The animals were kept under standard laboratory conditions (22 ± 2 °C, 60% relative humidity, and 12 h light/12 h dark cycle) and had ad libitum access to standard rodent feed (PK-120-1, Laboratorsnab, Ltd., Moscow, Russia) and water. To assess the influence of oral IQ-1S administration on the progression of the AMD-like pathology, we randomly assigned 4-month-old male OXYS rats to one of two groups (10 rats per group). One group was given intragastric administration of 50 mg of IQ-1S per kilogram of body weight in 2 mL of 1% starch mucus for 45 days (from age 4.5 to 6 months). The second group of OXYS rats received only starch mucus. As controls, we also used a group of 10 Wistar rats (parent strain, healthy rats). All experimental procedures were in compliance with the European Communities Council Directive of 24 November 1986 (86/609/EEC). The animal study was approved by the Commission on Bioethics of the Siberian State Medical University (Protocol No. 4 008/4/06/2022 dated 20 June 2022). Every effort was made to minimize the number of animals used and their discomfort. The IQ-1S-treated and untreated OXYS rats and Wistar rats were euthanized by an overdose of isoflurane inhalation anesthesia; the posterior wall of both eyes were carefully removed.

### 2.2. IQ-1S

The sodium salt of 11*H*-indeno[1,2-*b*]quinoxalin-11-one oxime (IQ-1S) (M314 series) was synthesized as described previously [19]. The chemical structure of IQ-1S was confirmed by the methods of mass spectrometry and nuclear magnetic resonance; sample purity was 99.9%. To prepare the IQ-1S emulsion, a weighted amount of IQ-1S powder, corresponding to the proper dose for the animal, was aseptically pounded with a pestle with 20 μL of Tween 80; 2.0 mL of physiologic sodium chloride solution was then added to create a suspension.

### 2.3. Histological Examination

For histological and immunohistochemical assays, the samples were fixed in 10% neutral formaldehyde in 0.1 mol/L phosphate buffer (pH 7.4) and embedded in paraffin according to the standard method [35]. Serial frontal sections (4 to 5 µm thick) were made, stained with H&E, and examined with a photomicroscope (Axiostar Plus, Carl Zeiss, Germany). Morphometric parameters were measured by quantitative image analysis performed with Axiovision software, version 4.8 (Zeiss, Thornwood, NY, USA). Evaluation was performed by examining five sections of the retina for each animal at a magnification of 10 × 100, using a frame area of 900 μm^2^. The specific area of the choroidal vessels (open, with stasis, aggregation of blood cells, or thrombosis), and the specific area of the RPE layer, were measured, and their ratio to the total area of the choroid or retina, respectively, was calculated. Separately, the number of ganglionic neurons with central and total chromatolysis, and nuclear pyknosis per 200 corresponding cells of the layer, was counted. The percentage of photoreceptors with nuclear pyknosis per 1000 photoreceptors, the number of layers in 10 fields of view of each retinal slice, and the distribution density of the photoreceptor nuclei in the eyepiece frame of 900 μm^2^ were calculated, these were then recalculated for an area of 1 mm^2^ of the slice; the percentage of radial glial cells and neurons in the inner nuclear and ganglionic layers per 200 corresponding retinal cells was calculated.

### 2.4. Immunohistochemical Examination (Retinal Immunohistochemistry Staining)

Slides containing intact tissue were placed in a thermostat at 37 °C for 30 minutes. The slide was then simultaneously deparaffinized by placing xylene-xylene-100% ethanol-100% ethanol-95% ethanol-90% ethanol-80% ethanol-70% ethanol. After rinsing, the slides were immersed in 3% H_2_O_2_ for 10 min. Then buffer with citric acid was added to the glass slide and heated to boiling. The sections were blocked with 5% BSA and then incubated at 37 °C for half an hour. The sections were then incubated with primary antibodies to VEGF (Fine Biotech Co., Ltd., Wuhan, China), PEDF (Abcam, Waltham, MA, USA), and p53 (Novus Biologicals, Centennial, CA, USA), colored at 4 °C overnight. Then, after incubation for 30 min at +25 °C with the secondary antibody, the sections were washed in two portions of the buffer solution. To detect staining, sections were incubated with an aqueous solution of chromogen (AEC Chromogen/Substrate Kit, Novus Biologicals, USA) for 10 min at +25 °C, after which they were washed with two portions of tap water and counterstained with Mayer’s hematoxylin. The immunohistochemical reaction was assessed as positive when reddish-brown staining was detected in the cytoplasm or on the membrane of the studied cells. Five rats were used and five sections of each rat were counted. The amount of all immunopositive structures as a percentage of the total number of structures in the retinal section was calculated.

### 2.5. Electron Microscopic Examination

Retinal samples of untreated and IQ-1S treated OXYS rats and Wistar rats (*n* = 5 per group) were fixed with 2.5% glutaraldehyde in 0.1 M sodium cacodylate buffer (pH 7.2) for 1 h, washed with 0.1 M sodium cacodylate buffer, and post-fixed in 1% osmium tetroxide in the same buffer for 1 h. After that, the samples were washed with water and incubated in a 1% aqueous solution of uranyl acetate in the dark at RT for 1 h. The samples were then dehydrated using a graduated series of ethanol and acetone mixtures, and embedded with a mixture of epon-araldite resins. First, semi-thin sections, 1 μm thick, were made on an ultratome and stained with toluidine blue, then ultra-thin sections were made. Ultra-thin sections were stained with uranyl acetate and lead citrate, and then examined under a transmission electron microscope (JEM 100 SX; Jeol, Tokyo, Japan) at the Interdepartmental Joint Center for Microscopic Analysis of Biological Objects, Institute of Cytology and Genetics, Siberian State University, Department of the Russian Academy of Sciences. On electron micrographs of the associative and ganglionic layers of the retina (45 photos per group of animals), all organelles located in these areas were stained using the software Adobe Photoshop. For each photograph, the following parameter was determined: the specific total area of each type of organelle located in the electron-transparent areas of neurons.

### 2.6. Statistics

Statistical analysis was carried out using the Statistica 10 software package (Statsoft, St Tulsa, OK, USA), implementing the methods of variation statistics. To assess the significance of differences when comparing mean values, the nonparametric Mann–Whitney test was used. The data are presented as median and interquartile range 25th and 75th percentiles (Me (Q25%–Q75%)) or as the mean± standard deviation of the mean (SD). Differences were considered statistically significant at *p* < 0.05.

## 3. Results

### 3.1. IQ-1S Attenuates Pathological Changes in the Retina of OXYS Rats

Consistent with our previous observations, by the age of 6 months there were alterations in all layers of the untreated OXYS rats’ retinas. Thus, aberrations of RPE cells flattening, with a variable size and shape of their nuclei, typical to AMD, as well as some decrease in the thickness of the photoreceptor layer were identified (Table 1). In the RPE cells, we found pathological changes in the ultrastructure typical for the OXYS rats: disintegration of microvilli, disappearance of basal striation, accumulation of a large number of phagolysosomes, lipofuscin, and the presence of individual cells with marginal condensation of chromatin in the nucleus, which is one of the signs of the development of apoptosis (Figure 1).

The degree of damage to photoreceptor cells in the untreated OXYS rats was significant. Photoreceptor cells died mainly due to necrosis, with the development of nuclear pyknosis, perikaryon edema, cell lysis, and proliferation of scleral processes in Muller cells, as evidenced by a significant decrease in the density of the photoreceptor cell nuclei and an increase in the percentage of pyknosis among this population (Figure 2, Table 1). 

The degree of damage to neurons in the inner layers of the retinas of the OXYS rats was lower than in the photoreceptor layer. At the same time, both in the inner nuclear and ganglionic layers of the retinas of the OXYS rats, the frequency of apoptosis was higher than in the Wistar rats, as evidenced by an increase in cells with signs of pyknosis. Increased apoptosis in ganglionic neurons, and in the inner nuclear layer, was confirmed by the presence of cells positive for the apoptosis-related marker, p53 protein, which were not identified in the outer nuclear layer (Figure 3, Table 1).

These aberrations occur against the background of significant changes in the vessels of the choroid. Along with vessels with normal blood supply, we observed the appearance of a certain number of capillaries with signs of partial occlusion. In contrast to the choroids of Wistar rats, in the choroids of the OXYS rats, blood flow disturbances were revealed: the aggregation of blood cells, the stasis, and the thrombosis of small vessels. As a result, the specific area of open functional vessels in the OXYS rats was 2.7 times less, and the number of vessels with sludge, erythrocyte stasis, and thrombosis was 13 times greater than in the Wistar rats (*p* < 0.05). An ultrastructural study confirmed the development of obliteration of the choroidal vessels, and revealed the destruction of the connective tissue components of the Bruch’s membrane in the form of an increase in osmiophilia and granular disintegration of collagen structures (Figure 4).

Treatment with IQ-1S significantly prevented thrombus formation in retinal vessels, which improved microcirculation and, consequently, led to the greater preservation of the RPE cells and photoreceptors. At the age of 6 months, the number of rows in the outer nuclear layer in the retinas of the OXYS rats (Table 1) was slightly, but significantly, less than in the Wistar rats. At the same time, the numerical density of photoreceptor nuclei was 1.5-fold less. The IQ-1S treatment did not affect the thickness of the photoreceptor layer and increased the density of photoreceptors; however, it remained somewhat lower than in Wistar rats. Importantly, the treatment with IQ-1S reduced the proportion of photoreceptors with signs of apoptosis approximately threefold, and also significantly reduced the proportion of the p53+ neurons in the ganglion layer and at the trend level in the inner nuclear layer. Additionally, IQ-1S improved the condition of the Bruch’s membrane. However, electron microscopic examination showed that the majority of animals which received IQ-1S still had a decrease in the Bruch’s membrane thickness and increased osmiophilia. Besides, RPE cells in IQ-1S-treated OXYS rats remained more flattened than in Wistar rats, but the presence of microvilli on their surface indicated that the cells were functionally active (Table 1, Figure 1).

Next, we assessed the influence of IQ-1S treatment on the ultrastructural state of the associative and ganglion neurons in the retinas of the OXYS rats. At the age of 6 months, substantial changes in the structural organization of cellular organelles were observed in the neurons of untreated OXYS rats in comparison with the Wistar rats (Figure 5, Table 2).

An electron microscopic study (Table 2) revealed a significant decrease in the number of mitochondria in the retinal neurons: the specific area of mitochondria in the cytoplasm of associative and ganglion neurons in 6-month-old OXYS rats was less than in the Wistar rats (*p* < 0.01 for both). Treatment with IQ-1S significantly increased this parameter (*p* < 0.03 for both cell types); however, the specific area of mitochondria in the cytoplasm of both associative and ganglionic neurons in the retinas of the OXYS rats remained significantly lower than in the Wistar rats (*p* < 0.03 and *p* < 0.001, respectively). Both associative and ganglion neurons of the OXYS rats have a smaller specific area of rough endoplasmic reticulum cisterns (ER, *p* < 0.01 for both). At the same time, mitochondria swell, the cristae are destroyed in them, the cisterns of the endoplasmic reticulum segregate, lose ribosomes, expand, which leads to the formation of vacuoles and is accompanied by an increase in the specific area of vacuoles in both associative and ganglionic neurons in the OXYS rats (*p* < 0.02 and *p* < 0.01, respectively). Treatment with IQ-1S increased the specific area of the rough ER in both types of neurons (*p* < 0.03), but only in associative neurons reached the level of the Wistar rats, while in ganglionic neurons it remained half as much (*p* < 0.01). Both associative and ganglion neurons in the retinas of the OXYS rats did not have significant differences in the specific area of the Golgi apparatus and lysosomes compared to the Wistar rats. Treatment with IQ-1S also did not affect the Golgi apparatus, however, it led to a decrease in the specific areas of vacuoles in both the associative and ganglion cells (*p* < 0.03 for both), but the indicator remained higher in the OXYS rats than in the Wistar rats.

### 3.2. IQ-1S Improved the Expression of VEGF and PEDF in the Retina of OXYS Rats

At the next stage, we evaluated the effect of IQ-1S on the expression of key regulators of angiogenesis, the main pro-angiogenic VEGF, and the most potent natural angiogenesis inhibitor and PEDF (Figure 6). VEGF immunohistochemical staining was weak in choroidal vessels in all animal groups, but was slightly elevated in the control OXYS rats and was not detectable in the IQ-1S treated OXYS rats. At the same time, VEGF immunoreactivity is pronounced in intraretinal vessels, where VEGF-positive cells were twice as many in the OXYS rats compared to the Wistar rats (*p* < 0.001). Treatment with IQ-1S significantly reduced this parameter (*p* < 0.029), but it still remained higher than in the Wistar rats (*p* < 0.028). VEGF immunoreactivity was also strongly detected in ganglion neurons, where the number of VEGF+ cells was twice as high in the OXYS rats as in the Wistar rats and was significantly reduced by IQ-1S treatment (*p* < 0.01 for both), and, as a result, did not differ from that in Wistar rats. At the same time, immunohistochemical staining did not reveal VEGF+ cells among associative neurons of all groups of rats.

We did not find PEDF+ cells in both the choroid and intraretinal vessels. PEDF immunoreactivity was clearly observed in the ganglion and association neuron cells in rat retinas, with PEDF+ cell density in both types of neurons being significantly lower in the OXYS rats than in the Wistar rats (*p* < 0.01). IQ-1S treatment increased the proportion of PEDF+ cells among both types of neurons (*p* < 0.01 for both).

## 4. Discussion

The aim of this study was to evaluate the retinoprotective potential of the JNK signaling pathway inhibitor IQ-1S using OXYS rats as an AMD model. As we have shown earlier, the clinical signs of AMD-like pathology against the background of structural and functional changes in RPE cells, and disorders of choroidal microcirculation, develop in 100% of the eyes of OXYS rats by the age of ~3–4 months [29,30,36]. Thus, it can be argued that at the time of the start of the treatment with the drug, the OXYS rats already had the corresponding clinical and pathomorphological signs of the disease. Here we have shown that at the age of 6 months, the main morphological signs of AMD-like pathology were expressed well in the retina of the OXYS rats from the control group. IQ-1S treatment started at 4 months of age significantly suppressed the progression of AMD-like pathology in the OXYS rats.

Retinopathy in the OXYS rats, consistent with dry AMD in humans, was characterized by progressive RPE degeneration and choroidal involution with secondary loss of photoreceptors. We have previously shown that destruction of RPE cells is the primary change in the development of retinopathy in OXYS rats, and begins as early as 20 days of age. It is manifested by a decrease in the area of RPE cells, an increase in the proportion of multinucleated cells, and a violation of their hexagonal shape. The progression of an AMD-like pathology in OXYS rats is accompanied by the excessive accumulation of beta-amyloid (Aβ) and lipofuscin in RPE cells, as well as a violation of cell morphology, their hypertrophy, and reactive gliosis [30,31,37]. In the present study, using light and electron microscopy, typical pathological changes in the morphology and ultrastructure of the RPE in 6-month-old OXYS rats were detected in the form of flattening of aberrational cells and destruction of microvilli. Treatment with IQ-1S significantly improved the structural and functional parameters of RPE. In treated animals, we observed a high preservation of cytoplasmic organelles, including the basal labyrinth and microvilli, which is a sign of the functional activity of the RPE. The specific area of the RPE was significantly higher than in the group without treatment, but did not reach the Wistar rat values.

There is growing evidence that changes in the choroid and choroidal microcirculation that occur with age may play a critical role in the pathogenesis of AMD, and localized choroidal dysfunction and abnormal hemodynamics are an important disease mechanism for AMD [38,39]. The manifestation of clinical signs of AMD in OXYS rats at the age of three months occurs against the background of age-related changes in blood flow in the choroidal vessels [30]. Here we have shown that pronounced signs of partial occlusion were observed in both choroidal and intraretinal vessels of 6-month-old OXYS rats. The violation of hemodynamics in the vessels of the choroid was accompanied by changes in the vascular wall, a fibrosis, and a violation of the structure of the Bruch’s membrane, which was manifested by increased osmiophilia and granular breakdown of collagen structures. These data are consistent with the data of Zarbin M.A. [40]: under conditions of retinal aging, changes occur in the functioning of the Bruch’s membrane proteins, accumulation of collagen glycosylation products, which leads to disruption of the RPE functioning, its destruction, and the death of photoreceptors. Treatment with IQ-1S significantly improved the state of retinal vessels, led to the preservation of the three-layer structure of the Bruch’s membrane, which contributed to greater preservation of neurons in the retinas of OXYS rats. At the same time, some thinning of the membrane and an increase in its electron density were still observed, and blood flow disturbances in the choroid of OXYS rats persisted: the number of open choriocapillaries was reduced, and the number of choriocapillaries with signs of occlusion was increased compared to Wistar rats. 

It is hypothesized that the JNK pathway may play a key role in the development of AMD due to its role in stress responses and involvement in apoptosis, inflammation, and VEGF production. JNK1 is a critical factor in hypoxia-induced retinal VEGF production and may contribute to hypoxia-induced pathological angiogenesis [12]. We have previously shown that the first signs of retinopathy in OXYS rats develop upon suppression of both proangiogenic VEGF and antiangiogenic neurotrophic PEDF proteins [36,41]. In the present study we revealed signs of higher expression of VEGF in intraretinal and choroidal vessels, as evidenced by an increase in the proportion of VEGF-positive cells. Nonetheless, we did not reveal signs of neovascularization in the retinas of 6-month-old untreated OXYS rats. Treatment with IQ-1S significantly decreased the proportion of the VEGF-positive endothelial cells, to the level of Wistar rats.

PEDF is multimodal and has neurotrophic, anti-angiogenic, anti-inflammatory, and antioxidant properties that can protect the cells of the inner retina and retinal ganglion cell layer from death caused by ischemia and cytotoxic agents [42]. Although there are a lot of data on the inhibition of VEGF production via the JNK-dependent pathway in the literature, we did not find data on the effect of JNK1 inhibitors on the expression of PEDF, as well as convincing data on its regulation via the JNK signaling pathway. At the same time, IQ-1S treatment not only reduced the number of VEGF-positive endothelial cells in retinal vessels in OXYS rats, but also increased the percentage of PEDF-positive associative and ganglion neurons, which was significantly reduced in untreated OXYS rats. Perhaps this effect of the drug is achieved by improving blood flow, reducing retinal tissue hypoxia, which also mediates the neuroprotective activity of IQ-1S.

Destructive changes in the RPE can lead to an increase in the death of photoreceptors, which according to morphological manifestations corresponds to necrosis, and is observed in the retina of untreated OXYS rats. At the same time, in the neurons of the inner layers of the retina of untreated OXYS rats, we observed an increase in perikaryon edema and a significant destruction of organelles. In addition, immunohistochemically in the neurons of this region of the retina of the control OXYS rats, we registered the activation of the p53-dependent apoptosis pathway. There are data in the literature on the role of JNK3 in p53 phosphorylation [43], which was indirectly confirmed in our study, since treatment with the JNK3 inhibitor IQ-1S significantly reduced the number of p53-positive associative and ganglion neurons in the retinas of OXYS rats, thus contributing to their survival.

There is growing evidence that mitochondrial dysfunction plays a critical role in the pathogenesis of AMD [44]. Mitochondrial dysfunction was considered the most likely cause of accelerated aging in OXYS rats. It was first identified in the liver [45], then in the muscles [46], the myocardium [47], and recently in brain structures [48]. Here we showed for the first time that the development of an AMD-like pathology in OXYS rats occurs against the background of disturbances in the ultrastructure of mitochondria and a significant decrease in their number in the neural retina: the specific area of mitochondria in the cytoplasm of associative and ganglionic neurons in 6-month-old OXYS rats was significantly less than in Wistar rats. IQ-1S treatment improved mitochondrial ultrastructure and increased their number.

Along with a decrease in the specific area of mitochondria in OXYS rats at the age of 6 months, a significant decrease in the specific area of granular ER was revealed, while cisternae preserved in the cytoplasm of neurons underwent pronounced ultrastructural changes: loss of ribosomes from the membrane surface, and segregation of cisternae with the formation of small cavities that expanded and served as a site for the formation of vacuoles. The observed changes are characteristic of a phenomenon called granular EPR stress [49]. At the same time, ER stress itself can stimulate cell death by modulating JNK activity [50]. IQ-1S reduced the manifestations of ER stress, which, together with the limitation of mitochondrial damage, led to a decrease in vacuolization and contributed to slowing down the destruction of neurons in the inner layers of the retina. 

Collectively, our data support that the increase of the JNK-signaling pathway may play an important role in the pathophysiology of AMD. It seems probable that use of the JNK3 inhibitor IQ-1S can be a good prophylactic strategy to maintain retinal health and to treat AMD. However, for this, it is necessary to conduct a detailed study of the mechanisms of IQ-1S action.

## Figures and Tables

**Figure 1 biomedicines-11-00395-f001:**
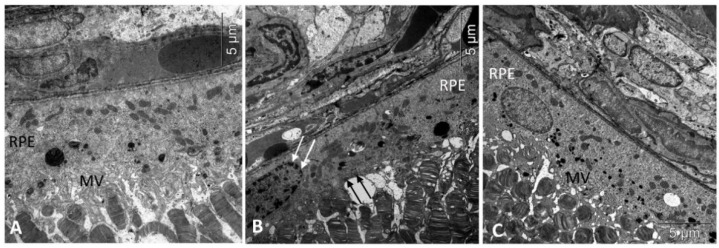
Treatment with IQ-1S improved the structural and functional parameters of RPE in 6-month-old OXYS rats. Electron micrographs show examples of normal ultrastructural signs of RPE in Wistar rats (**A**), untreated (**B**), and IQ-1S treated (**C**) OXYS rats. RPE images from untreated OXYS rats (**B**) show evidence of RPE cell disorganization-nuclear chromatin condensation (arrow) and microvilli destruction (black arrows) in OXYS rats. RPE cells from OXYS rats treated with IQ-1S (**C**) have normal nuclear and cytoplasmic ultrastructure. The presence of microvilli (MV) on their surface indicates that the cells are functionally active.

**Figure 2 biomedicines-11-00395-f002:**
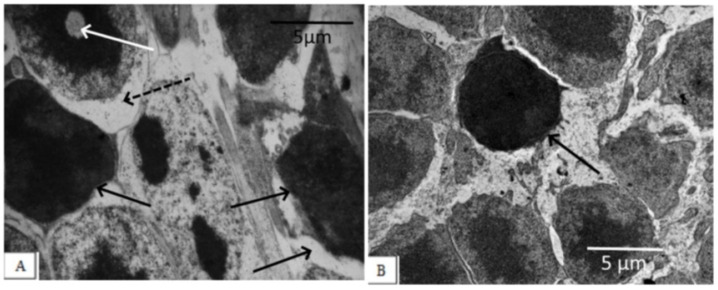
Outer nuclear layer of OXYS rats without correction (**A**) and on the background of IQ-1S treatment (**B**). In OXYS rats, pronounced changes in the nuclear part of the photoreceptors were observed: nuclear pyknosis (black arrows), perikaryon edema (dashed arrows), and nuclear lysis (white arrow). On the background of treatment with IQ-1S, the edema phenomena leveled out, a greater preservation of photoreceptor nuclei was observed. However, pyknosis phenomena took place.

**Figure 3 biomedicines-11-00395-f003:**
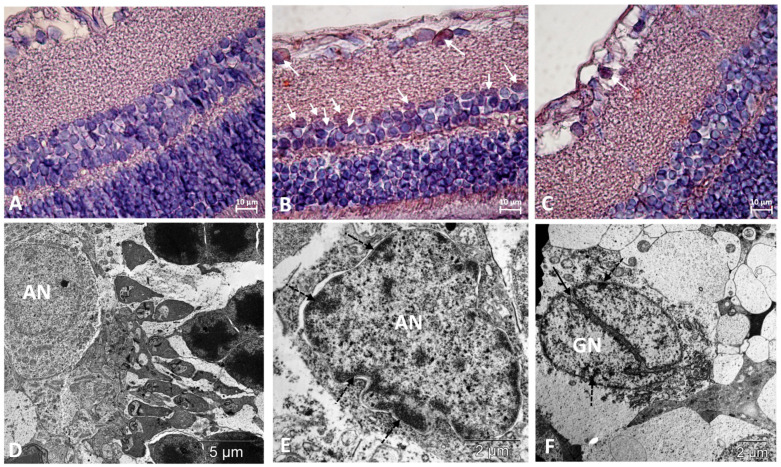
Immunostaining did not reveal p53-positive cells in the retinas of Wistar rats (**A**) and revealed them among neurons in the ganglionic and inner nuclear layers of the retina of OXYS rats (white arrows) (**B**). Treatment with IQ-1S significantly decreased the number of p53-positive cells in the retinas of OXYS rats (**C**). The electron micrographs show ultrastructural features of the normal structure of an associative neuron (AN) in Wistar rats (**D**) and examples of morphological manifestations of apoptosis in retinal neurons of OXYS rats: (**E**) nuclear invagination and marginal chromatin condensation (dashed arrows) in the associative neuron; (**F**) division of the nucleus of the ganglionic neuron (GN), marginal condensation of chromatin in the nucleus (dashed arrows). (**A**–**C**): representative images of immunohistochemistry. Additional staining with hematoxylin. (**D**–**F**): electron diffraction pattern.

**Figure 4 biomedicines-11-00395-f004:**
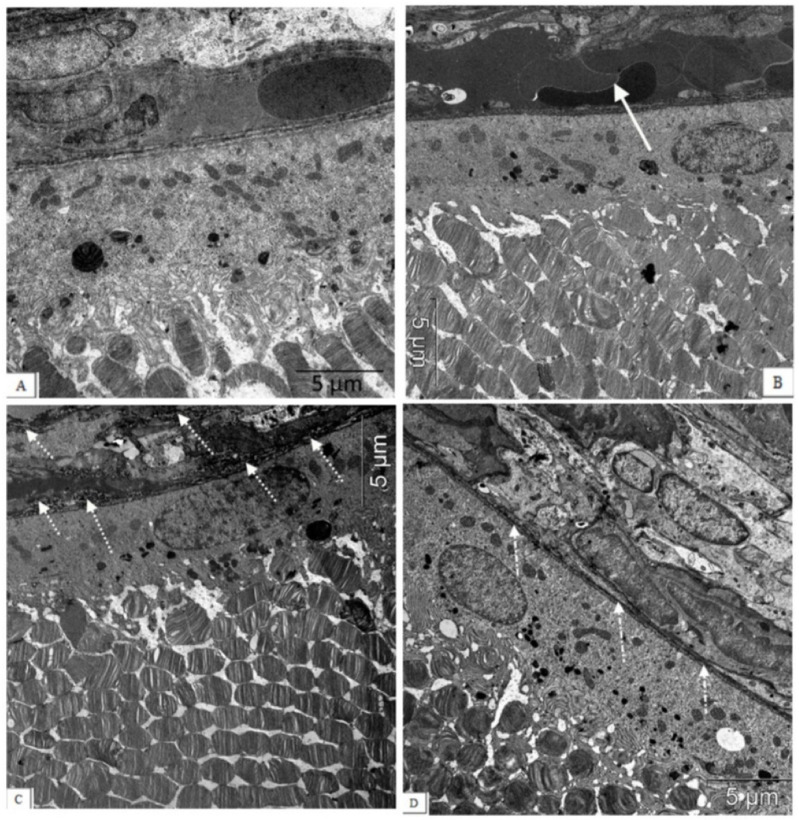
The electron micrographs show examples of normal ultrastructural features of the chorioretinal complex of a Wistar rat (**A**), untreated (**B**,**C**) and an IQ-1S-treated (**D**) OXYS rat. Stasis, sludge of the blood vessels of the OXYS rat’s choroid (white arrow), destructive changes in the structure of the Bruch’s membrane, and capillary endothelium (dashed arrow) are presented. During treatment with IQ-1S, the structure of the Bruch’s membrane was intact, however, its thinning and increased osmiophilia were sometimes observed ((**D**), white dot-dotted arrow).

**Figure 5 biomedicines-11-00395-f005:**
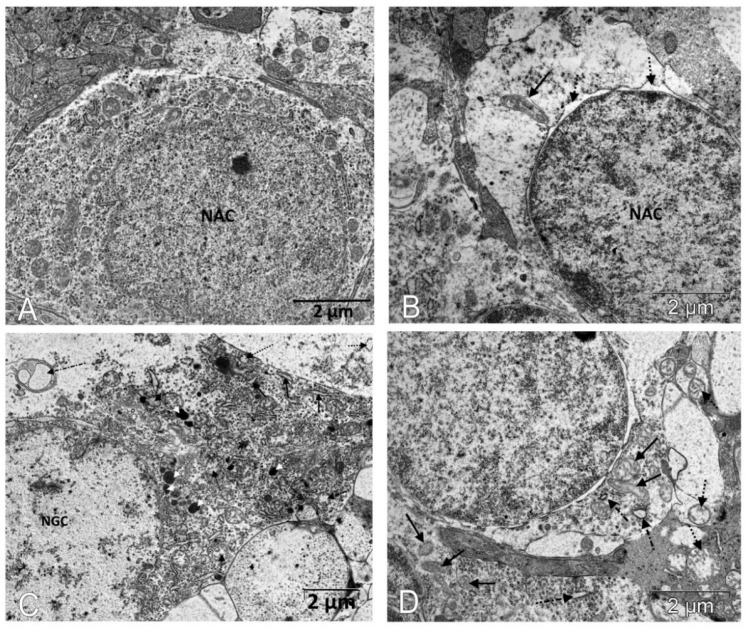
IQ-1S treatment improved the ultrastructure of neurons of the inner layers of the retinas in 6-month-old OXYS rats. The electron micrographs show examples of the ultrastructure of neurons of the inner layers of the retina of Wistar rats (**A**), untreated OXYS rats (**B**,**C**), and IQ-1S-treated OXYS rats (**D**). In the cytoplasm of the associative retinal neurons of untreated OXYS rats, there is an expansion of the perinuclear space (dashed arrows), destruction of most organelles, an increase in the size of preserved mitochondria, which have external signs of damage to the inner membrane of the cristae - local formation of electron-dense granules, tubular cristae (black arrow). (**B**). A focal destruction of organelles, a vacuolization, and a formation of multivesicular bodies (dot-dotted arrow) occur in most of the ganglion neurons; an expansion of granular ER cisterns (black arrows), an increase in the number of lysosomes (white dashed arrows), and a destruction of mitochondria are observed in the preserved parts of the cytoplasm (black dashed arrows) in the retinas of untreated OXYS rats (**C**). Treatment with IQ-1S leads to a greater preservation of organelles, including mitochondria (black arrows), but some of them increase in size; swelling and destruction of cristae are observed in them (dashed arrow). The cisterns of the granular ER are fragmented and expand, but do not lose their connection with the ribosomes (dot-dotted arrow) (**D**).

**Figure 6 biomedicines-11-00395-f006:**
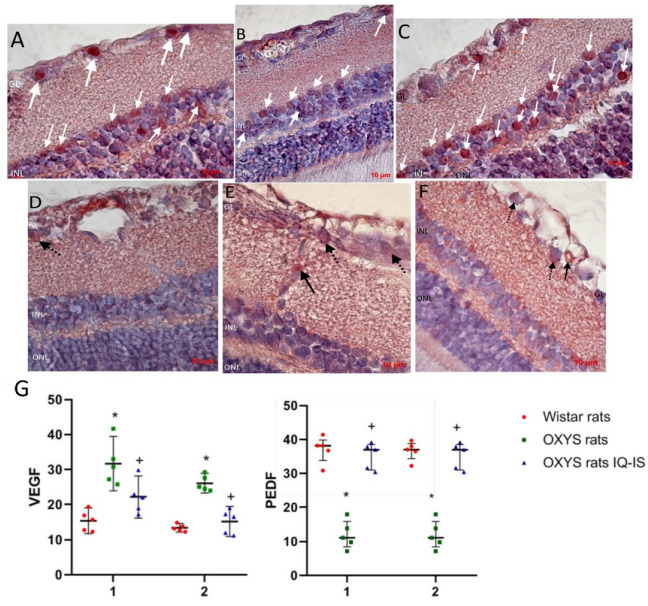
Detection and quantitative analysis of elements immunopositive to VEGF (**D**–**F**) and PEDF (**A**–**C**) in the retinas of Wistar rats (**A**,**D**), untreated OXYS rats (**B**,**E**), and IQ-1S-treated OXYS rats (**C**,**F**). Ganglionic neurons of the ganglionic layer (GL) and associative neurons of the inner nuclear layer (INL) were immunoreactive to PEDF in the neuroretina (white arrows). Vascular endothelium (black arrows) and ganglionic neurons (dashed arrows) were VEGF-immunopositive structures. IQ-1S treatment improved PEDF (**C**) and VEGF (**F**) expression in the retina of OXYS rats. Expression of VEGF in intraretinal vessels (1) and ganglion neurons (2) of Wistar rats, control rats, and OXYS rats treated with IQ-1S (**G**). The data are presented as the amount of VEGF- and PEDF-positive cells as a percentage of the total number of structures studied in the entire area of the retinal section (M ± SD).

**Table 1 biomedicines-11-00395-t001:** Effects of IQ-1S on the morphometric parameters of the Retina of OXYS Rats.

Parameters	Wistar	OXYS	OXYS + IQ-1S
Specific area of Open choriocapillaris, %	32.9 (24.6; 39.0)	12,1 (10.4; 15.0) *	20.0 (15.6; 20.8) *^+^
Specific area of choriocapillaris with stasis and thrombosis, %	1.71 (1.26; 1.86)	22.23 (15.59; 20.81) *	10.0 (6.3; 11.0) *^+^
Specific area of the layer RPE, %	6.57 (6.28; 7.30)	3,12 (2.99; 3.17) *	4.83 (4.53; 5.19) *^+^
Photoreceptors with nuclear pyknosis, %	0.40 (0.32; 0.40)	23.5 (17.5; 24.3) *	8.3 (7.8; 8.8) *^+^
Number of rows of photoreceptor nuclei	12.0 (11.3; 13.0)	10.0 (10.3; 11.8) *	10.0 (9.0; 10.8) *
Numerical density of photoreceptors, nuclei per mm^2^	39,505 (35,022; 40,943)	24,935 (24,935; 26,485) *	32,958 (26,873; 33,918) *^+^
p53 immunopositive neurons in the ganglion layer, %	0	6.00 (3.42; 8.2) *	1.67 (1.00; 2.00) *^+^
p53 immunopositive neurons in the inner nuclear layer, %	0	4.47 (4.0; 7.0)	2.00 (1.94; 2.6)
p53 immunopositive neurons in the outer nuclear layer, %	0	0	0

*—significant differences (*p* ˂ 0.05) between the strains; ^+^—a significant effect (*p* ˂ 0.05) of IQ-1S within the strain. Data are presented as median and interquartile range—25th and 75th percentiles (Me (Q25%–Q75%)). IQ-1S was given at 50 mg per day from 4.5 to 6 months.

**Table 2 biomedicines-11-00395-t002:** Effects of IQ-1S on the specific areas of organelles in the cytoplasm of associative and ganglion neurons (according to electron micrographs).

Organelles	Type of neurons	Wistar	OXYS	OXYS + IQ-1S
Rough ER, %	Associative	20.41 (17.59; 29.72)	8.67 (3.89; 8.86) *	16.22 (12.69; 16.69) ^+^
	Ganglion	43.85 (41.17; 59.28)	13.5 (13.09; 15.69) *	21.50 (16.24; 26.09) *^+^
Mitochondria, %	Associative	16.29 (16.13; 16.66)	3.51 (3.19; 3.79) *	9.23 (7.02; 14.43) *^+^
	Ganglion	16.98 (16.72; 17.49)	3.55 (2.26; 5.21) *	7.41 (6.53; 7.80) *^+^
Golgi apparatus, %	Associative	1.26 (1.18; 1.39)	0.64 (0.53; 0.83)	1.75 (1.16; 2.06)
	Ganglion	1.86 (1.43; 2.93)	1.66 (0.79; 1.77)	1.41 (1.09; 2.92)
Lysosomes, %	Associative	0.49 (0.35; 0.73)	0.31 (0.27; 0.37)	0.39 (0.02; 0.77)
	Ganglion	0.58 (0.55; 1.02)	0.56 (0.27; 0.80)	0.64 (0.02; 1.39)
Vacuoles, %	Associative	0.08 (0.06; 0.1)	2.40 (1.73; 2.4) *	1.00 (0.8; 1.09) *^+^
	Ganglion	0.19 (0.04; 0.36)	3.18 (2.37; 3.4) *	1.12 (0.98; 1.16) *^+^

*—significant differences (*p* ˂ 0.05) between the strains; ^+^—a significant effect (*p* ˂ 0.05) of IQ-1S within the strain. Data are presented as median and interquartile range—25th and 75th percentiles (Me (Q25%–Q75%)). IQ-1S was given at 50 mg per day from 4.5 to 6 months.

## Data Availability

Raw data are available from the corresponding author upon request.

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
