# Peer review of "Suppression of Age-Related Macular Degeneration-like Pathology by c-Jun N-Terminal Kinase Inhibitor IQ-1S"

_biomedicines, 2023, doi:10.3390/biomedicines11020395_

Round 1

Reviewer 1 Report

In the manuscript entitled “Suppression of Age-Related Macular Degeneration-Like Pathology by c-Jun N-Terminal Kinase Inhibitor IQ-1S”, the authors evaluate the effect of c-Jun inhibitor IQ-1S on AMD-like retinal structural in the senescence-accelerated OXYS rats. They found that, IQ-1S could improve choroidal and retinal blood circulation, increase RPE functional, decrease photoreceptors and RGCs degeneration. They also describe changes of retinal ultrastructure and retina VEGF, PEDF positive cell numbers after IQ-1S treatment. The results are interesting. This study may provide evidence of retinoprotective potential of IQ-1S in AMD-like pathology development. But there are still some issues need to be addressed.

1. In Table 2, all neuron types were Ganglion cell. Did the authors investigate the organelles in RPEs,photoreceptors or other retinal neurons after IQ-1S treatment?

2.  The authors investigate changes of retina VEGF, PEDF positive cell numbers after IQ-1S treatment. It can’t be described as “IQ-1S improved the expression of VEGF and PEDF in the retina”. It well be better to evaluate retina VEGF, PEDF expression by qPCR and WB.

3.  In Figure 1, there was no dashed arrow in panel B. Please make sure the figure legend was consistent with the illustration. Arrows in most electron micrographs were not obvious. Please make them clearer.

4. In Figure 2, did the “External nuclear layer” mean outer nuclear layer?

5. In immunohistochemistry images, retina should be shown in the same direction, eg. INL upwards.

Reviewer 2 Report

Figure 1 legend B says there is a dashed arrow, but the photo only has black arrows and white arrows.

Also, figure 1 findings are not convincing when looking at the photo and reading the description in the legend.

Figure 5 legend refers to dot-dotted arrow but should say dashed.

The legends and labeling of the photos are confusing, making it even more challenging to try to interpret the results in the figures.

Overall, the imaging results showing benefit with treatment are difficult to confirm.

Round 2

Reviewer 2 Report

Line 274 should say dashed instead of dot-dotted.